

# Emptiness instanton in quantum polytropic gas

**Alexander G. Abanov[1] and Dimitri M. Gangardt[2*]**

**1** Department of Physics and Astronomy,
Stony Brook University, Stony Brook, NY 11794, USA
**2** School of Physics and Astronomy, University of Birmingham,
Edgbaston, Birmingham, B15 2TT, UK

⋆ d.m.gangardt@bham.ac.uk

## Abstract

The emptiness formation problem is addressed for a one-dimensional quantum polytropic gas characterized by an arbitrary polytropic index $\gamma$, which defines the equation of state $P \sim \rho^\gamma$, where $P$ is the pressure and $\rho$ is the density. The problem involves determining the probability of the spontaneous formation of an empty interval in the ground state of the gas. In the limit of a macroscopically large interval, this probability is dominated by an instanton configuration. By solving the hydrodynamic equations in imaginary time, we derive the analytic form of the emptiness instanton. This solution is expressed as an integral representation analogous to those used for correlation functions in Conformal Field Theory. Prominent features of the spatiotemporal profile of the instanton are obtained directly from this representation.



# 1 Introduction

Rare fluctuations leading to large deviations in many-body systems have attracted significant attention [1–7] in recent years. This interest is partially driven by precise measurements of particle number fluctuations in ultracold quantum gases [8–14]. Among these, the emptiness formation probability (EFP) stands out as one of the most iconic and extensively investigated examples. In certain integrable models, the EFP can be analyzed using the Bethe ansatz and serves as a benchmark for assessing the reliability of approximate non-perturbative methods, such as instanton calculus. The latter relies on the fact that, in the limit of a large number of particles, the main contribution to the partition function comes from a single macroscopic configuration of fields. Such a configuration often exhibits sharp boundaries separating fluctuating regions.

The appearance of a nonrandom boundary separating fluctuating regions in random systems is known as the limit shape phenomenon. These limit shapes emerge in statistical mechanics systems that are sensitive to boundary conditions. Notable examples include the formation of arctic curves in dimer systems, emptiness formations in quantum gases and polymers, and the equilibrium shapes of quantum crystals (for a review and references, see [15, 16]). Most known examples where analytical treatments are available can be mapped to one-dimensional free fermion models, with rare exceptions, such as works on six-vertex [17–21] and four- and five-vertex models [22, 23] with domain wall boundary conditions.

In Ref. [24], a hydrodynamic approach was developed to address the problem of finding large fluctuations in fluid systems. The method originated from the conventional bosonization technique [25] and is based on solving fluid dynamics equations in imaginary time. This approach has the potential to be applied to many limit shape problems in both interacting and non-interacting systems. However, finding analytical solutions to fluid dynamics equations for a generic equation of state remains challenging.

Recently, progress has been made in this direction, effectively adding another example of an interacting model amenable to analytical solutions for limit shape problems. In Ref. [3], the authors showed that the emptiness formation problem can be solved within the hydrodynamic approach for a one-dimensional polytropic gas with a power-law equation of state $P \sim \rho^\gamma$, where $P$ and $\rho$ are the pressure and the density of the gas at zero temperature. They constructed the solution for an infinite discrete sequence of values of the polytropic index:

$$\gamma = 1 + \frac{2}{2n+1}, \tag{1}$$

where $n$ is an integer. This includes some important cases: $n = 0$, $\gamma = 3$ for free fermions in 1D and $n = 2$, $\gamma = 7/5$ for free fermions confined to quasi-1D geometry by a transverse harmonic trap [26]. Based on this solution, they conjectured the result for the probability of emptiness formation for arbitrary values of $\gamma$.

The emptiness formation is an example of a rare fluctuation (instanton) with an exponentially suppressed probability [24]. Indeed, for an empty interval of length $R$ to appear in a compressible liquid, one must create a spacetime disturbance of a typical size $R \times R/c_0$, where $c_0$ is the thermodynamic sound velocity. The Emptiness Formation Probability (EFP) is then given by the probability of such a disturbance appearing spontaneously,

$$\mathcal{P}_{\text{EFP}} \sim e^{-\frac{R^2 \rho_0 m c_0}{\hbar} f(n)}. \tag{2}$$

Here, $m$ is the mass of the particles and $\hbar$ is the Planck's constant. One recognizes the dimensionless spacetime area, defined using the microscopic length and time intervals $1/\rho_0$ and $\hbar/mc_0^2$. The dimensionless area is assumed to be large for the instanton approach to be valid

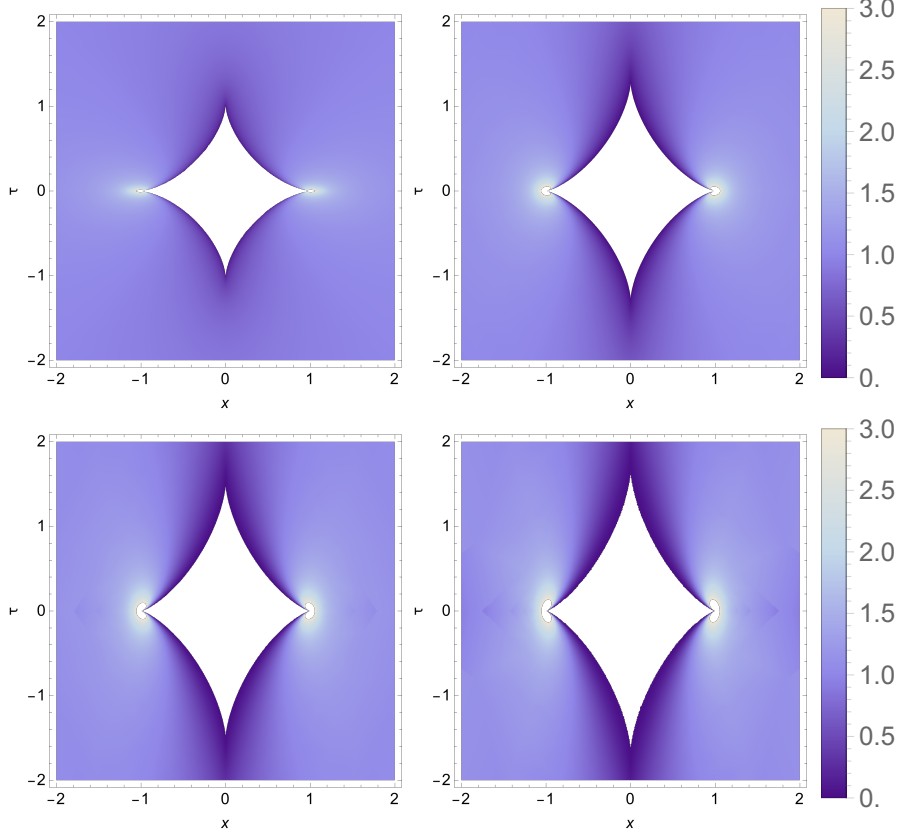

Figure 1: Emptiness instanton spatiotemporal density profiles for various polytropic indices, both integer and non-integer. From left to right: upper row, $n = 0$, $n = 0.5$, lower row, $n = 1$, $n = \sqrt{2}$. Color scheme for the density $\rho(x,\tau)$ is shown on the right.

and serves as the large parameter of the semiclassical approach. The dimensionless factor

$$f(n) = \frac{2}{n+1}\left(\frac{\Gamma(n+3/2)}{\Gamma(n+1)}\right)^2, \tag{3}$$

was obtained in Ref. [3] for integer values of $n$ from the emptiness instanton solutions.

However, generally speaking, $n$ does not have to be an integer. A well-known example is provided by a weakly interacting Bose gas, for which the mean-field equation of state has $\gamma = 2$, corresponding to $n = 1/2$.[1] In this case, the best approach so far has been based on the numerical solution of the corresponding hydrodynamic equations, as in the earlier work, Ref. [2].

In this work, our goal is to extend the hydrodynamic approach to arbitrary values of the polytropic index by analytically continuing emptiness instanton configurations away from integer values of $n$. Our main result is the analytic solution for the spatiotemporal density profile shown in Fig.1 for $\gamma$ corresponding to various, not necessarily integer, values of $n$. In particular, we consider $n = 1/2$, $\gamma = 2$, a case studied in Ref. [2]. The key feature is the formation of an empty spacetime region that includes the interval $-1 < x < 1$ at $\tau = 0$ and extends into a finite-time region, persisting up to a critical imaginary time $\pm\tau_c$. The critical time $\tau_c$ is given by

$$\tau_c = \frac{\Gamma(n+3/2)}{\Gamma(3/2)\Gamma(n+1)}. \tag{4}$$

---

[1]Nor does $n$ have to be positive: the case of $n = -1$, $\gamma = -1$ corresponds to the Chaplygin gas, which has recently gained considerable interest in cosmology [27]. More generally, this applies to $-1 \le \gamma < 0$.

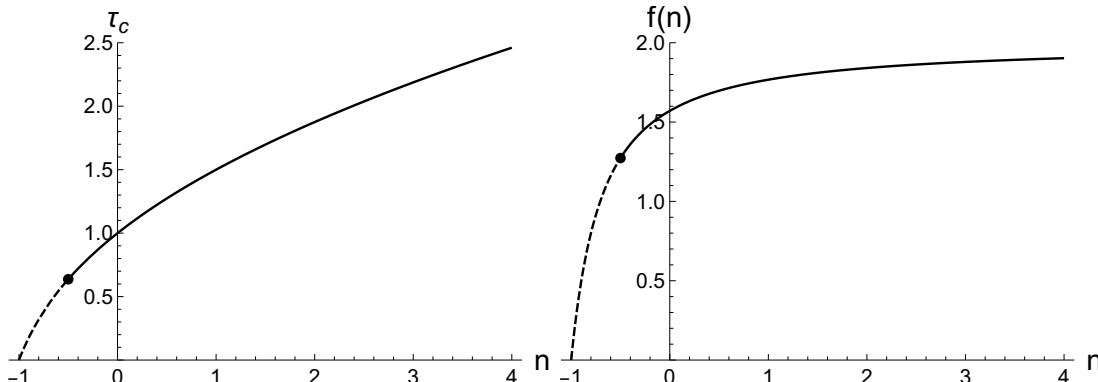

Figure 2: Left: Temporal extension of the emptiness instanton, $\tau_c$, as a function of $n$, given by Eq. (4). Right: Dimensionless action $f(n)$ of the emptiness instanton, Eq. (3). For free fermions, $f(0) = \pi/2$, and in the limit $n \to \infty$, it approaches the asymptotic value $f(\infty) = 2$. While our results are valid for $n > -1/2$ (solid curve), both $\tau_c$ and $f(n)$ can be analytically continued as shown by dashed curves to the case of the Chaplygin gas, $n = -1$, where $\tau_c = f(n) = 0$.

Its dependence on $n$ is shown in Fig. 2, along with $f(n)$. The emptiness region generalizes the astroid shape found in Ref. [24] for the free-fermion case $n = 0$. For $x, \tau$ values away from this region, the density approaches its equilibrium value, $\rho \to 1$.

The paper is organized as follows. For completeness and to establish notation, we briefly review the hydrodynamic approach to the EFP in Section 2 and summarize the results of Ref. [3], which use standard hydrodynamics continued to imaginary time, in Section 3. In Section 4, we perform the analytic continuation to arbitrary values of $n$ and independently verify that the corresponding solution satisfies the correct boundary conditions, thereby representing the emptiness instanton. In particular, we rederive for general $n$ the leading asymptotics of the emptiness probability, Eqs.(2) and (3), reported in Ref. [3] for integer $n$ and conjectured there to hold for arbitrary values of the polytropic index. The spatiotemporal properties of this solution are studied numerically in Section 5. In the concluding Section 6, we comment on our method and propose possible extensions.

## 2 Emptiness formation probability for polytropic gas

Parametrizing density and current density in terms of displacement field $u(x, t)$

$$\rho = \partial_x u, \qquad j = \rho v = -\partial_t u,$$

the Emptiness Formation Probability (EFP) is given by the integral over configurations

$$\mathcal{P}_{\text{EFP}}(R) = \frac{1}{Z} \int_{\text{empt}}' \mathcal{D}u \, e^{\frac{i}{\hbar}S[u]}, \tag{5}$$

restricted by the emptiness condition,

$$\rho(x, 0) = \partial_x u(x, 0) = 0, \qquad |x| < R. \tag{6}$$

The normalization factor $Z$ is given by the unrestricted integral over all configurations,

$$Z = \int_{\text{all}} \mathcal{D}u \, e^{\frac{i}{\hbar}S[u]}. \tag{7}$$

The configurations' contribution is controlled by the action

$$S[u] = S[\partial_x u, \partial_t u] = \iint dt\,dx \left[ \frac{mj^2}{2\rho} - V(\rho) \right], \tag{8}$$

where the integration domain consists of the whole space-time, $x \in [-\infty, \infty]$, $t \in [-\infty, \infty]$. The integrand in the action, Eq. (8), is the difference of kinetic energy density, $mj^2/2\rho$ and internal energy density $V(\rho)$, given by

$$V(\rho) = \frac{mc_0^2 \rho_0}{\gamma - 1} \left[ \frac{1}{\gamma} \left( \frac{\rho}{\rho_0} \right)^\gamma - \frac{\rho}{\rho_0} \right]. \tag{9}$$

The internal energy density includes the chemical potential term, linear in density, that fixes the thermodynamic density $\rho_0$ and corresponds to the power-law or *polytropic* equation of state,

$$P(\rho) = \frac{mc_0^2 \rho_0}{\gamma} \left( \frac{\rho}{\rho_0} \right)^\gamma, \tag{10}$$

for pressure $P = \rho^2 \partial_\rho(V/\rho)$ and the density characterised by the polytropic index.

For the density deviating from this value, the sound velocity is given by the thermodynamic relation,

$$c = \sqrt{\frac{1}{m} \frac{\partial P}{\partial \rho}} = c_0 \left( \frac{\rho}{\rho_0} \right)^{\frac{\gamma - 1}{2}}. \tag{11}$$

To ensure that uniform density $\rho_0$ represents a stable thermodynamic configuration we need $\gamma > 1$ and we restrict our analysis to these values of polytropic index.[2]

For sufficiently large $R$, the integrals in the definition of EFP, Eq. (5) can be calculated semiclassically by minimizing the action. This procedure amounts to solving hydrodynamic equations of motion with the emptiness boundary condition (6) for $t = 0$ in addition to $u(x, t) = u_0(x, t) = \rho_0 x$ for $x, t \to \infty$. As explained in Ref. [3] such solution can only be achieved by Wick rotation, $t \to -i\tau$ in Eq. (8). Using dimensionless coordinates and fields,

$$x \to Rx, \qquad \tau \to T\tau, \qquad \rho = \rho_0 \rho, \qquad v \to (R/T)v, \tag{12}$$

with $T = (R/c_0) \times (\gamma - 1)/2$ we obtain the corresponding Euclidean action

$$\frac{i}{\hbar} S[\partial_x u, \partial_\tau u] = -\frac{2}{(\gamma - 1)} \frac{R^2 \rho_0 mc_0}{\hbar} \iint d\tau\,dx \left[ \frac{j^2}{2\rho} + \frac{\gamma - 1}{4} \left( \frac{\rho^\gamma}{\gamma} - \rho \right) \right]. \tag{13}$$

The prefactor is large and provides justification for the semiclassical calculation.

As a consequence of our choice of the time unit $T$, the dimensionless internal energy gets an extra factor $(\gamma - 1)^2/4$ compared to Eqs. (12),(13) of Ref. [3]. The same factor now appears in the imaginary time Euler equation,

$$\partial_\tau v + v \partial_x v = \frac{(\gamma - 1)^2}{4} \rho^{\gamma - 2} \partial_x \rho. \tag{14}$$

Solving this equation together with the continuity equation

$$\partial_\tau \rho + \partial_x(\rho v) = 0, \tag{15}$$

---

[2]The marginal value $\gamma = 1$ corresponds to $n \to \infty$ and requires $V(\rho) = mc_0^2 \rho \log(\rho/\rho_0)$.

subject to emptiness boundary condition Eq. (6) gives the *Emptiness Instanton* solution $u^*(x, \tau)$. Once this solution is found, EFP is given by

$$\mathcal{P}_{\text{EFP}}(R) \simeq e^{\frac{i}{\hbar} S_{\text{inst}}}, \tag{16}$$

where the instanton action, $S_{\text{inst}} = S[u^*] - S[u_0]$ represents the difference of the action evaluated on the emptiness instanton solution and the action evaluated on $u_0 = \rho_0 x = x$ corresponding to the flat background solution dominating the normalization integral, Eq. (7). As explained after Eq. (8) the configurations of the field $u(x, \tau)$ are defined on the whole imaginary time axis, $\tau \in [-\infty, \infty]$ and obey the time-reversal symmetry, $u(x, -\tau) = u(x, \tau)$. This allows one to re-express the instanton action as double the contribution of the outward trajectory, $\tau \in [0, \infty]$ as it was done in [3]. Also, by virtue of Wick rotation, $iS_{\text{inst}}/\hbar$ is real and negative, as expected in instanton calculus leading to exponentially small EFP.

## 3 Emptiness instanton for integer $n$

In Ref. [3] the emptiness instanton configurations for the density and velocity fields $\rho(x, \tau)$, $v(x, \tau)$ were obtained for special values of polytropic index corresponding to integer nonnegative values of $n$, see Eq. (1). This result was based on the standard hydrodynamic approach outlined in Appendix A adapted to imaginary time evolution. Below we present it in the form suitable for analytic continuation to non-integer values of $n$.

In terms of Riemann invariants $\lambda$ and characteristic velocities $w$[3,4]

$$\lambda = v + (2n+1)ic = v + i\rho^{\frac{1}{2n+1}}, \tag{17}$$

$$w = v + ic = v + \frac{i}{2n+1}\rho^{\frac{1}{2n+1}}, \tag{18}$$

the equations of motion (14,15) become

$$\partial_\tau \lambda + w(\lambda, \bar{\lambda})\partial_x \lambda = 0, \tag{19}$$

and its complex conjugate. Then the emptiness instanton configuration is given implicitly as the solution of the "ballistic propagation" equation

$$x - w\tau = \partial_\lambda \mathcal{V}_n(\lambda, \bar{\lambda}), \tag{20}$$

and its complex conjugate. The main ingredient of this solution is the potential $\mathcal{V}_n(\lambda, \bar{\lambda})$ which must satisfy the Euler-Poisson equation,

$$\partial_{\bar{\lambda}}\partial_\lambda \mathcal{V}_n = n\frac{\partial_\lambda \mathcal{V}_n - \partial_{\bar{\lambda}} \mathcal{V}_n}{\lambda - \bar{\lambda}}. \tag{21}$$

Eq. (21) must be supplemented by boundary conditions appropriate for the emptiness formation problem. As the emptiness interval is fixed to be $\tau = 0$, $|x| < 1$ we might expect singularities at the end points of this interval. By analogy with the case of integer $n$ [3] we anticipate divergence of density and velocity fields at $x = \pm 1$, which is equivalent to

$$\partial_\lambda \mathcal{V}_n\Big|_{|\lambda| \to \infty} = \pm 1. \tag{22}$$

---

[3] Our choice of time unit $T$ was motivated by modified definitions of $\lambda, w$ given in Eq. (17),(18) making the behavior at space-time infinity $\text{Im}\,\lambda \to 1$ independent of $\gamma$.

[4] For polytropic gas we have a linear relation $(2n+1)w(\lambda, \bar{\lambda}) = (n+1)\lambda + n\bar{\lambda}$ between Riemann invariants and characteristic velocities. This fact allows one to find a closed solution for the polytropic gas emptiness instanton.

At spacetime infinity $x, \tau \to \infty$ we will have $v, \rho - 1 \to 0$. Solving linearized equations in this limit we obtain the leading (quadrupole) asymptotics

$$v + \frac{i}{2n+1}(\rho - 1) \sim \frac{i\alpha}{2n+1} \left( x - \frac{i}{2n+1}\tau \right)^{-2}. \tag{23}$$

In particular, for $\tau = 0, x \to \infty$ we have $v = 0$ and $\rho - 1 \sim \alpha/x^2$, equivalent to

$$\partial_\lambda \mathcal{V}_n \Big|_{\lambda = -\bar{\lambda} = i\mu \to i} \sim \sqrt{\frac{\alpha}{2n+1} \frac{1}{\mu - 1}}. \tag{24}$$

It was shown in Appendix B of Ref. [3] that the constant $\alpha$ determines the emptiness formation probability Eq. (2) via

$$f(n) = 2\pi \frac{2n+1}{2n+2} \alpha. \tag{25}$$

Finding the analytical form of the potential $\mathcal{V}_n$ satisfying Eqs. (21),(22),(24) is central to our work.

For the case of free fermions, $n = 0, \gamma = 3$, the potential is well known,

$$V_0(\lambda, \bar{\lambda}) = \sqrt{\lambda^2 + 1} + c.c., \tag{26}$$

see Ref. [24]. For a positive integer $n$ it was shown in Ref. [3] that the potential was systematically constructed in the form of the sum

$$\mathcal{V}_n(\lambda, \bar{\lambda}) = \sum_{m=0}^{n-1} \frac{a_m F_m(\lambda)}{(\lambda - \bar{\lambda})^{n+m}} + c.c., \tag{27}$$

with the coefficients $a_m$ and functions $F_m(\lambda)$ obeying the following recurrence relations:

$$a_m = -\frac{(n-m)(n+m-1)}{m} a_{m-1}, \qquad a_0 = 1, \tag{28}$$

$$F_{m-1}(\lambda) = \partial_\lambda F_m(\lambda), \qquad F_{n-1}(\lambda) = \frac{1}{n!} \lambda(\lambda^2 + 1)^{n-\frac{1}{2}}. \tag{29}$$

The goal of the present work is to construct the potential for non-integer values of $n$, thus extending the solution of the emptiness problem to arbitrary values of the polytropic index. As a first step, we note that by using the explicit expressions

$$a_m = \frac{(-1)^m}{m!} \frac{(n+m-1)!}{(n-m-1)!}, \qquad F_m(\lambda) = \partial_\lambda^{n-1-m} F_{n-1}(\lambda), \tag{30}$$

the sum (27) can be represented in a rather compact way as a multiple derivative,

$$\mathcal{V}_n(\lambda, \bar{\lambda}) = \frac{1}{n!} \partial_\lambda^{n-1} \frac{\lambda(\lambda^2 + 1)^{n-\frac{1}{2}}}{(\lambda - \bar{\lambda})^n} + c.c. \tag{31}$$

This representation is still limited to positive integer values of $n$. The crucial observation leading to the results in this paper is that for these values of $n$ Eq. (31) can be identified with the residue evaluation of the following contour integral:

$$\mathcal{V}_n(\lambda, \bar{\lambda}) = \frac{1}{n} \oint_{\mathcal{C}(\lambda, \bar{\lambda})} \frac{dz}{2\pi i} \frac{z(z^2 + 1)^{n-\frac{1}{2}}}{(z - \lambda)^n (z - \bar{\lambda})^n}, \tag{32}$$

provided the contour $\mathcal{C}(\lambda, \bar{\lambda})$ shown in Fig. 3 encircles the $n$-th order poles at $\lambda, \bar{\lambda}$ in the positive direction. This integral representation is the starting point of the analytical continuation of emptiness instanton solution away from the positive integers.

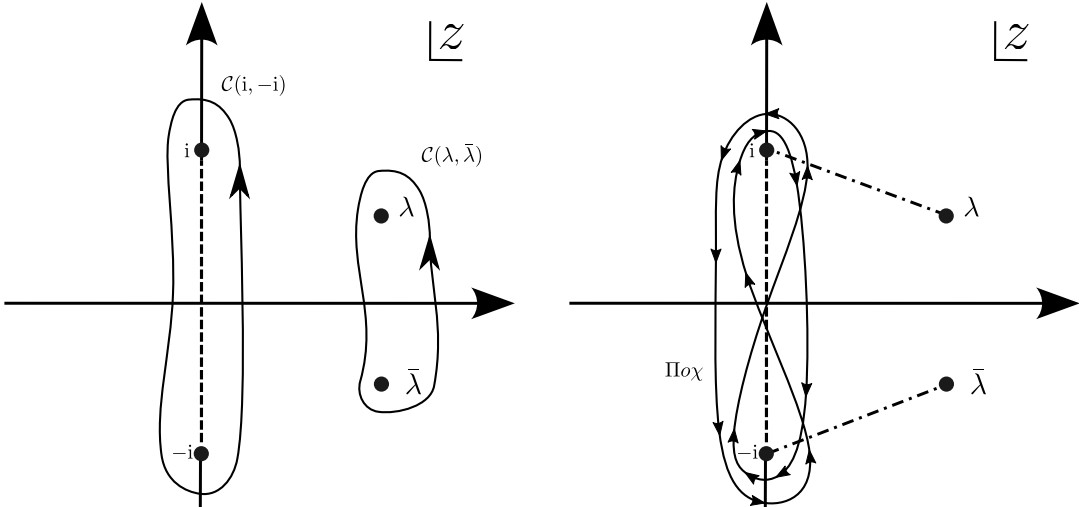

Figure 3: Left: contours $\mathcal{C}(\lambda,\bar{\lambda})$ and $\mathcal{C}(\mathrm{i},-\mathrm{i})$ in Eqs (32) and (33). Right: Pochhammer contour $\Pi o \chi$ in Eqs (34) and (35).

## 4 Emptiness instanton for arbitrary $n$

The integral representation of the potential, Eq. (32) is instrumental as it ensures that the Euler-Poisson equation, Eq. (21) is automatically satisfied. Indeed, Eq. (21) holds for the function $(z-\lambda)^{-n}(z-\bar{\lambda})^{-n}$ for any $z$ and therefore it holds for the whole integral by linearity.

By expanding the integration contour and taking into account the contribution at infinity the potential can be represented as

$$\mathcal{V}_n(\lambda,\bar{\lambda}) = \lambda + \bar{\lambda} - \frac{1}{n} \oint_{\mathcal{C}(\mathrm{i},-\mathrm{i})} \frac{\mathrm{d}z}{2\pi\mathrm{i}} \frac{z}{\sqrt{z^2+1}} \left(\frac{z-\mathrm{i}}{z-\lambda}\right)^n \left(\frac{z+\mathrm{i}}{z-\bar{\lambda}}\right)^n, \tag{33}$$

with the contour of integration $\mathcal{C}(\mathrm{i},-\mathrm{i})$ encircling the square root branch cut between $\mathrm{i}$ and $-\mathrm{i}$, see left panel in Fig. 3.

To extend this construction to a non-integer $n$ one has to modify the integration contour as for a general $n$ there will be additional branch cuts connecting points $\mathrm{i}, \lambda$ and $-\mathrm{i}, \bar{\lambda}$ as shown in the right panel of Fig. 3. Going around the branch points will end up on a different Riemann sheet. To avoid this we employ the standard procedure, similar to the classic treatment of beta function (see *e.g.* [28]) consisting in using the Pochhammer contour $\Pi o \chi$ which starts at an arbitrary point in the interval $-\mathrm{i}, \mathrm{i}$, goes around $\mathrm{i}, -\mathrm{i}$ in the positive (counter-clockwise) sense, then encircles $\mathrm{i}$ and $-\mathrm{i}$ in the negative sense before closing, see right panel of Fig. 3,

$$\mathcal{V}_n(\lambda,\bar{\lambda}) = \lambda + \bar{\lambda} - \frac{2}{(1+e^{2\pi\mathrm{i}n})^2} \frac{1}{n} \oint_{\Pi o \chi} \frac{\mathrm{d}z}{2\pi\mathrm{i}} \frac{z}{\sqrt{z^2+1}} \left(\frac{z-\mathrm{i}}{z-\lambda}\right)^n \left(\frac{z+\mathrm{i}}{z-\bar{\lambda}}\right)^n. \tag{34}$$

At the starting point, e.g., $z = 0$, the arguments of $z \pm \mathrm{i}$ are $\pm\pi/2$ respectively, while the arguments of $z - \lambda$ and $z - \bar{\lambda}$ lie in the interval $[-\pi, \pi]$. The additional $n$-dependent factor in front of the integral in Eq. (34) ensures the condition $\partial_\lambda \mathcal{V}_n(0,0) = \partial_{\bar{\lambda}} \mathcal{V}_n(0,0) = 0$ for any $n$ which follows from the parity symmetry. We prove this fact in Appendix B. For integer $n$, this factor equals $1/2$, compensating the Pochhammer contour containing two copies of the contour $\mathcal{C}(\mathrm{i},-\mathrm{i})$.

The r.h.s. of Eq. (20) is thus given by

$$\partial_\lambda \mathcal{V}_n(\lambda,\bar{\lambda}) = 1 - \frac{2}{(1+e^{2\pi\mathrm{i}n})^2} \oint_{\Pi o \chi} \frac{\mathrm{d}z}{2\pi\mathrm{i}} \frac{z}{\sqrt{z^2+1}} \left(\frac{z-\mathrm{i}}{z-\lambda}\right)^n \left(\frac{z+\mathrm{i}}{z-\bar{\lambda}}\right)^n \frac{1}{z-\lambda}. \tag{35}$$

It remains to show that Eq. (35) leads to a valid solution to the emptiness problem, *i.e.* it satisfies the conditions (22) and (24). In order to do this, it is convenient to manipulate the integral Eq. (35) by changing the variable $z = \mathrm{i}s$ and making the integration contour to collapse onto an interval of real axis:

$$\partial_\lambda \mathcal{V}_n(\lambda, \bar{\lambda}) = 1 - \frac{1}{\pi} \int_{-1}^{1} \frac{s\,\mathrm{d}s}{\sqrt{1-s^2}} \left(\frac{s-1}{s+\mathrm{i}\lambda}\right)^n \left(\frac{s+1}{s+\mathrm{i}\bar{\lambda}}\right)^n \frac{1}{s+\mathrm{i}\lambda}. \tag{36}$$

Note that $n$-dependent prefactor in Eq. (35) is compensated by the phase of the integrand on different segments of the collapsed contour $\Pi o \chi$.

The condition (22) follows straightforwardly from this integral representation for $n > -1/2$ corresponding to $\gamma > 1$. To demonstrate (24) we write for $\lambda = -\bar{\lambda} = \mathrm{i}\mu$

$$\partial_\lambda \mathcal{V}_n(\mathrm{i}\mu, -\mathrm{i}\mu) = 1 + \frac{1}{\pi} \int_{-1}^{1} \frac{(1-s^2)^{n-\frac{1}{2}}(s^2 + \mu s)}{(\mu^2 - s^2)^{n+1}}\,\mathrm{d}s. \tag{37}$$

The antisymmetric term proportional to $\mu$ in the numerator does not contribute to the integral by symmetry. The integral of the symmetric part can be expressed as a hypergeometric function,

$$\partial_\lambda \mathcal{V}_n(\mathrm{i}\mu, -\mathrm{i}\mu) = 1 + \frac{2}{\pi} \int_{0}^{1} \frac{s^2(1-s^2)^{n-\frac{1}{2}}}{(\mu^2 - s^2)^{n+1}}\,\mathrm{d}s = 1 + \frac{\Gamma\left(\frac{3}{2}\right)\Gamma\left(n+\frac{1}{2}\right)}{\pi\Gamma(n+2)} \frac{{}_2F_1\left(n+1, \frac{3}{2}, n+2; \frac{1}{\mu^2}\right)}{\mu^{2n+2}}. \tag{38}$$

This expression has inverse square root singularity, Eq. (24), for $\mu \to 1^+$ as follows from the known expansion of a hypergeometric function, see Eq. 15.4.23 of [29]. The amplitude,

$$\alpha = \frac{1}{2(2n+1)} \left(\frac{\Gamma\left(n+\frac{3}{2}\right)}{\Gamma\left(\frac{3}{2}\right)\Gamma(n+1)}\right)^2, \tag{39}$$

is identical to that found in Ref. [3] for integer $n$,

$$\alpha = \frac{1}{2(2n+1)} \left[\frac{(2n+1)!!}{2^n n!}\right]^2, \tag{40}$$

since $\Gamma(n+3/2)/\Gamma(3/2) = 2^{-n}(2n+1)!!$. This coefficient determines the dimensionless action of emptiness instanton given by Eq. (25).

This completes our proof that Eqs. (34),(35) are indeed the correct generalisation of the solution for the emptiness problem to arbitrary values of polytropic index. The integrals in Eqs. (34),(35) are examples of Dotsenko-Fateev integrals that have been used for representing correlation functions in conformal field theory [30–32].

## 5 Spatiotemporal profile of the emptiness instanton

Having obtained the the potential $\mathcal{V}_n(\lambda, \bar{\lambda})$ we now have access to the density profile $\rho(x, \tau)$ for any value of polytropic index $\gamma(n)$ by solving Eq. (20) and its complex conjugate,

$$\tau = -\frac{2n+1}{\mathrm{Im}\,\lambda} \mathrm{Im}\,\partial_\lambda \mathcal{V}_n(\lambda, \bar{\lambda}), \tag{41}$$

$$x = \mathrm{Re}\,\mathcal{V}_n(\lambda, \bar{\lambda}) + \tau\,\mathrm{Re}\,\lambda, \tag{42}$$

for $x(\lambda, \bar{\lambda})$, $\tau(\lambda, \bar{\lambda})$ and using $\rho(\lambda, \bar{\lambda}) = (\mathrm{Im}\,\lambda)^{2n+1}$. Similarly one obtains the velocity profile $v(x, \tau) = \mathrm{Re}\,\lambda$. The spatiotemporal density profile is shown in Fig 1 for polytropic index $\gamma$ corresponding to arbitrary, not necessarily integer values of $n$.

For numerical evaluation of $\partial_\lambda \mathcal{V}_n(\lambda, \bar{\lambda})$ we use Eq. (36) and the alternative representation,

$$\partial_\lambda \mathcal{V}_n = \frac{1}{\pi i} \int_{-1}^{1} \frac{dq}{q} \frac{1}{\sqrt{1-q^2}} \left( \frac{1+q}{1-i\lambda q} \right)^n \left( \frac{1-q}{1-i\bar{\lambda}q} \right)^n \frac{1}{1-i\lambda q}, \tag{43}$$

derived in Appendix B. The different choice of fixed integration contours leading to the representations Eq. (36) and Eq. (43) results in different branch-cut structure of the potential $\mathcal{V}_n$ and its derivatives. This choice is dictated by the range of the densties one wishes to represent without spurious discontinuities and has to be tailored to a particular region in the $(x, \tau)$ plane one wishes to study. The representation Eq. (36) is suitable for the regions in which $\rho > 1$, like the vicinity of $\tau = 0$ axis, while the representation Eq. (43) is suitable for $\rho < 1$, like the vicinity of the axis $x = 0$.

The density profile can be found analytically on the whole line $x = 0$ as the integral Eq. (43) with $\lambda = -\bar{\lambda} = i\mu$ can be calculated in a closed from. Using it in Eq. (41) leads to

$$\tau = \frac{2n+1}{\pi} \int_{-1}^{1} dq \frac{(1-q^2)^{n-\frac{1}{2}}}{(1-\mu^2 q^2)^{n+1}} = \frac{\Gamma(n+3/2)}{\Gamma(3/2)\Gamma(n+1)} \frac{1}{\sqrt{1-\mu^2}}. \tag{44}$$

After inverting for $\rho = \mu^{2n+1}$ one obtains

$$\rho(0, \tau) = \left( 1 - \left( \frac{\tau_c}{\tau} \right)^2 \right)^{n+\frac{1}{2}}, \tag{45}$$

where $\tau_c$ is given by Eq. (4). In particular, near the points $x = 0$, $|\tau| \to \tau_c^+$, we have

$$\rho \sim \left( 2 \frac{|\tau| - \tau_c}{\tau_c} \right)^{n+\frac{1}{2}}. \tag{46}$$

Similarly, the density profile for $\tau = 0$ can be obtained by using the result for $\mu > 1$, Eq. (38), in Eq. (42). In the vicinity of the points $x = \pm 1$, $\tau = 0$ one can use asymptotics of hypergeometric functions to obtain

$$\rho \sim \left( \frac{A_n}{|x| - 1} \right)^{\frac{2n+1}{2n+2}}, \qquad A_n = \frac{\Gamma(n+1/2)}{2\sqrt{\pi}\Gamma(n+2)}. \tag{47}$$

The density profiles for $\rho(0, \tau)$ and $\rho(x, 0)$ are shown in Fig. 4.

To obtain the shape of the empty region we note that as one approaches its boundary the density vanishes (except for $x = \pm 1$, $\tau = 0$) so that the Riemann invariants become degenerate, $\lambda = \bar{\lambda} = \nu$. Substituting it into Eq. (20) leads to the one-dimensional parametrization,

$$x - \nu\tau = g(\nu), \tag{48}$$
$$\tau = -g'(\nu), \tag{49}$$

where the function on the right hand side is obtained from Eq. (43) and reads

$$g(\nu) = \partial_\lambda \mathcal{V}_n \Big|_{\lambda = \bar{\lambda} = \nu} = \frac{1}{\pi i} \int_{-1}^{1} \frac{dq}{q} \frac{(1-q^2)^{n-\frac{1}{2}}}{(1-i\nu q)^{2n+1}}. \tag{50}$$

Expanding the denominator for small $\nu$ and evaluating the integrals in terms of beta function gives $g(\nu) = \sum_{m=0}^{\infty} g_m \nu^{2m+1}$, where the coefficients

$$g_m = \frac{(-1)^m}{\pi} \frac{1}{2m+1} \frac{B(n+1/2, m+1/2)}{B(2n+1, 2m+1)} = \frac{1}{\sqrt{\pi}} \frac{(-1)^m}{m+1/2} \frac{\Gamma(n+m+3/2)}{\Gamma(n+1)\Gamma(m+1)}, \tag{51}$$

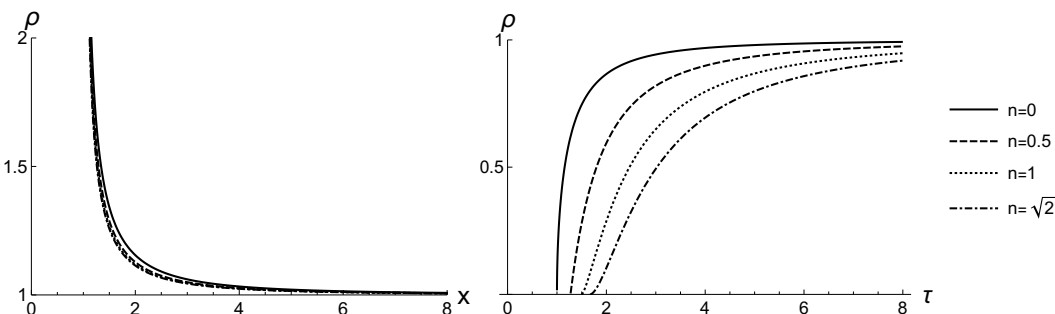

Figure 4: Left: Density profile at $\tau = 0$. Right: Density profile at $x = 0$ given by Eq. (45).

are identical to the Gauss series for the following combination of hypergeometric functions,

$$g(v) = \frac{\Gamma(n+1/2)}{\Gamma(n+1)\Gamma(1/2)} v \left[ {}_2F_1\left(n+1/2, 1/2, 1/2; -v^2\right) + 2n\, {}_2F_1\left(n+1/2, 1/2, 3/2; -v^2\right) \right]$$

$$= \frac{\Gamma(n+1/2)}{\Gamma(n+1)\Gamma(1/2)} v \left[ \frac{1}{(1+v^2)^{n+1/2}} + 2n\, {}_2F_1\left(n+1/2, 1/2, 3/2; -v^2\right) \right]. \tag{52}$$

Much simpler expression can be obtained for the derivative

$$g'(v) = \frac{\Gamma(n+3/2)}{\Gamma(3/2)\Gamma(n+1)} \frac{1}{(1+v^2)^{n+3/2}}. \tag{53}$$

These expressions can be used to study the shape of the emptiness boundary. Taking $v \to 0$ and expanding $g(v) \simeq v(g_0 + g_2 v^2)$ in Eqs. (48),(49) we obtain recover the value of the critical time

$$g_0 = g'(0) = \frac{\Gamma(n+3/2)}{\Gamma(3/2)\Gamma(n+1)} = \tau_c, \tag{54}$$

and the universal emptiness boundary shape, $|x| \sim (|\tau| - \tau_c)^{3/2}$, valid for any value of the polytropic index.

To study the shape of the emptiness region in the vicinity of the points $x = \pm 1$, $\tau = 0$ it is customary to use the transformation formulae of hypergeometric functions to write

$$g(v) = 1 + \frac{\Gamma(n+1/2)}{\Gamma(n+1)\Gamma(1/2)} \left[ \frac{v}{(v^2+1)^{n+1/2}} - \frac{1}{v^{2n}}\, {}_2F_1\left(n, n+1/2, n+1; -\frac{1}{v^2}\right) \right]. \tag{55}$$

Expanding this expression for $v \to \infty$ gives

$$g(v) \simeq 1 - \frac{\Gamma(n+3/2)}{\Gamma(1/2)\Gamma(n+2)} \frac{1}{v^{2n+2}}, \tag{56}$$

and we obtain $|\tau| \sim (1-|x|)^{\frac{2n+3}{2n+2}}$, which has a non-trivial dependence on the polytropic index. For $n = 0$ we recover the free-fermionic scaling exponent 3/2.

# 6 Conclusions and open questions

Our main achievement is determining the Emptiness Formation Probability (EFP) resulting from a large quantum fluctuation in a one-dimensional gas with a polytropic equation of state. Starting from the results of Ref. [3], obtained for the polytropic index $\gamma$, Eq. (1), with integer

$n$, we derived an elegant integral representation, Eqs. (34) and (36), for the potential $\mathcal{V}_n$ and its derivatives, which control the hydrodynamic solution in Eq. (20). This representation enables us to extend the solution of Ref. [3] to arbitrary values of the polytropic index, thus proving the conjectured EFP results, Eqs. (2) and (3). While the solution for the Emptiness Instanton we provide is valid for $\gamma > 1$, some of its parameters, like those shown in Fig 2., can be analytically continued down to Chaplygin gas case $\gamma = -1$. It is plausible that for these values our instanton configuration provides a meaningful solution to a problem which is different from the emptiness formation, and we wish to explore this possibility in the future.

The hydrodynamic method of finding instanton solutions, which we used here and which was introduced in Ref. [24], can be viewed as a nonlinear extension of the bosonization techniques previously used for the emptiness formation problem [25]. The advantage of this method is that it does not rely on the integrability of the underlying microscopic problem and can be generalized to non-integrable and higher-dimensional systems. However, there are only a few exact results for emptiness fluctuations (and, more generally, limit shapes) available in the literature beyond free-fermion models. The solution for the polytropic gas presented in this work and Ref. [3] is an important addition to that list.

In our computations, we started with the polytropic gas, a macroscopic model for an underlying system of interacting particles. One can, for example, consider one-dimensional particles interacting through a $1/r^{\gamma-1}$ potential. When coarse-grained, this potential results in the internal potential energy of the gas $\epsilon(\rho) \sim \rho^\gamma$. Additionally, for specific values of $\gamma$ or $n$, there are other realizations of the polytropic gas. For instance, the case $\gamma = 3$ ($n = 0$) corresponds to free quantum fermions, while $\gamma = 2$ ($n = 1/2$) can be realized as the hydrodynamic description of weakly interacting bosons, approximated by the nonlinear Schrödinger model.

Our findings enable us not only to compute the Emptiness Formation Probability (EFP) but also to determine the form of the hydrodynamic solution dominating the path integral for the emptiness formation process: the Emptiness Instanton. This solution is qualitatively similar to the one obtained in Ref. [24] for the free-fermion gas ($n = 0$) but exhibits $n$-dependent singularities in its spatiotemporal profile.

In particular, we find that the emptiness region has an astroid-like shape in spacetime, as shown in Fig.1, defined by the parametric form (49) with $g(v)$ given in (50). This parametric dependence can be understood as the Legendre transform from $v$ to $\tau$ of the velocity-coordinate distribution $x = g(v)$ at $\tau = 0$, as discussed in Ref. [33].

The emptiness boundary exhibits four cusp-like singularities. Near $x = 0, \tau = \pm\tau_c$, the behavior is controlled by universal free-fermionic exponents: $|x| \sim (|\tau| - \tau_c)^{3/2}$. In contrast, the scaling behavior at $\tau = 0, x = \pm1$ depends on $n$: $|\tau| \sim (1 - |x|)^{\frac{2n+3}{2n+2}}$. The density profile near these singularities is given by Eqs. (46) and (47) and has an $n$-dependent scaling form.

The question about fluctuations around the Emptiness Instanton solution arises naturally. We note that the polytropic gas equation of state with a generic $n$ cannot be obtained from fermions with short-range interactions, and one might look for non-Tracy-Widom fluctuations near the limit shape boundary. In particular, one may ask whether the dynamical KPZ exponent $3/2$, which describes the scaling near the limit shape boundaries, should be replaced by an $n$-dependent analogue.

The structure of our solution hints at an intriguing relationship between our hydrodynamic approach and Conformal Field Theory (CFT), where Dotsenko-Fateev integrals are known to represent the correlation functions of certain primary fields [30–32]. It is therefore tempting to associate the central ingredient of the hydrodynamic solution, the potential $\mathcal{V}_n$, with such correlation functions and to explore the structure of the corresponding CFT. CFTs describing limit shapes in inhomogeneous one-dimensional quantum systems for free fermions were recently proposed in Ref. [34,35]. Extending these methods beyond free fermions by considering the polytropic gas with $n \neq 0$ will be addressed in a separate publication.

Our work can be extended to other quantities dominated by macroscopic hydrodynamic configurations. One example is the full counting statistics (FCS) of particles on the segment $[-R, R]$ at time $\tau = 0$. For the free-fermion case ($n = 0$), this problem was solved using the hydrodynamic method in Ref. [6]. The similarity between the obtained FCS solution and the EFP solution from Ref. [24] suggests that this method can be extended to the interacting case ($n \neq 0$), and we are planning to explore this possibility.

## Acknowledgments

We are grateful to Alex Kamenev and Baruch Meerson for fruitful discussions, and to Ilya Gruzberg and Paul Krapivsky for carefully reading the manuscript. DMG gratefully acknowledges support from the Simons Center for Geometry and Physics at Stony Brook University, where part of this research was conducted.

**Funding information**    AGA's work was supported by the National Science Foundation under Grant NSF DMR–2116767.

## A   Hydrodynamics of polytropic gas

Hydrodynamic configurations are specified by density, $\rho(x, t)$, and velocity $v(x, t)$ fields obeying continuity equation,

$$\partial_t \rho + \partial_x (\rho v) = 0, \tag{A.1}$$

and Euler equation

$$\partial_t v + v \partial_x v + \frac{c^2}{\rho} \partial_x \rho = 0. \tag{A.2}$$

Following the standard procedure outlined in *e.g.* [36] these equations can be rewritten as

$$\partial_t \lambda_+ + w_+ \partial_x \lambda_+ = 0, \tag{A.3}$$
$$\partial_t \lambda_- + w_- \partial_x \lambda_- = 0, \tag{A.4}$$

in terms of characteristic velocities $w_\pm = v \pm c$ and the Riemann invariants

$$\lambda_\pm = v \pm \int^\rho \frac{c}{\rho} \, \mathrm{d}\rho.$$

To solve Eqs. (A.3),(A.4) one employs hodograph transform consisting of interchanging dependent and indpendent variables $(\lambda_+, \lambda_-)$ and $(x, t)$ by using the chain rule and its inverse:

$$\begin{pmatrix} \partial_+ \\ \partial_- \end{pmatrix} = \begin{pmatrix} \partial_+ x & \partial_+ t \\ \partial_- x & \partial_- t \end{pmatrix} \begin{pmatrix} \partial_x \\ \partial_t \end{pmatrix}, \qquad \begin{pmatrix} \partial_x \\ \partial_t \end{pmatrix} = \frac{1}{J} \begin{pmatrix} \partial_- t & -\partial_+ t \\ -\partial_- x & \partial_+ x \end{pmatrix} \begin{pmatrix} \partial_+ \\ \partial_- \end{pmatrix}, \tag{A.5}$$

where $\partial_\pm = \partial / \partial \lambda_\pm$ and the Jacobian $J = \partial_+ x \, \partial_- t - \partial_+ t \, \partial_- x$, is assumed to be nonzero. Eqs. (A.3),(A.4) become

$$\partial_- x - w_+ \partial_- t = 0, \tag{A.6}$$
$$\partial_+ x - w_- \partial_+ t = 0. \tag{A.7}$$

Up to now, the discussion has been completely general. In the case of polytropic sound velocity given by Eq. (11) one can go further by exploiting the linear relation between Riemann invariants $\lambda_\pm$ and characteristic velocities $w_\pm$,

$$w_\pm = \frac{1}{2}\left(1 \pm \frac{1}{2n+1}\right)\lambda_+ + \frac{1}{2}\left(1 \mp \frac{1}{2n+1}\right)\lambda_-\,. \tag{A.8}$$

As $\partial_+ w_- = \partial_- w_+$ it is possible to parametrize the solution with the help of real potential $\mathcal{V}$ as

$$x - w_\pm t = \partial_\pm \mathcal{V}_n\,. \tag{A.9}$$

Taking derivatives of this equation w.r.t. $\lambda_\pm$ and using (A.6),(A.7) one shows that this potential satisfies the Euler-Poisson equation

$$\partial_+ \partial_- \mathcal{V}_n = n\frac{\partial_+ \mathcal{V}_n - \partial_- \mathcal{V}_n}{\lambda_+ - \lambda_-}\,. \tag{A.10}$$

# B  Manipulating integrals

In this Appendix we present the details of the derivations of integral representations used in the main text.

Let us start by showing that (34) reduces to (32) for integer $n$. We consider

$$\mathcal{V}_n = \lambda + \bar{\lambda} - \frac{4}{(1+e^{2\pi i n})^2}I(n)\,, \tag{B.1}$$

where we introduce the following basic integral

$$I(n) = \frac{1}{2n}\oint_{\Pi o\chi}\frac{\mathrm{d}z}{2\pi \mathrm{i}}\frac{z}{\sqrt{z^2+1}}\left(\frac{z^2+1}{(z-\lambda)(z-\bar{\lambda})}\right)^n\,. \tag{B.2}$$

The contours of integrations as defined in Figure 3. The Pochhammer contour $\Pi o\chi$ starts at $z = 0$, goes around $\mathrm{i}$, $-\mathrm{i}$ in the positive (counter-clockwise) sense, then encircles $\mathrm{i}$ and $-\mathrm{i}$ in the negative sense before closing, see right panel of Fig. 3. The phase of $n$-th power in the integrand is taken to be 0 at the beginning of the contour $z = 0$. We also assume that $\operatorname{Re}\lambda = \operatorname{Re}\bar{\lambda} = v > 0$.

Then one can derive the following relations for integer $n$:

$$I(n \in \mathbb{Z}) = \frac{1}{n}\oint_{\mathcal{C}(\mathrm{i},-\mathrm{i})}\frac{\mathrm{d}z}{2\pi \mathrm{i}}\frac{z}{\sqrt{z^2+1}}\left(\frac{z^2+1}{(z-\lambda)(z-\bar{\lambda})}\right)^n \tag{B.3}$$

$$= \lambda + \bar{\lambda} - \frac{1}{n}\oint_{\mathcal{C}(\lambda,\bar{\lambda})}\frac{\mathrm{d}z}{2\pi \mathrm{i}}\frac{z}{\sqrt{z^2+1}}\left(\frac{z^2+1}{(z-\lambda)(z-\bar{\lambda})}\right)^n\,. \tag{B.4}$$

For an arbitrary $n$ one can reduce the integral (B.2) to the integral over the interval $[-\mathrm{i}, \mathrm{i}]$ or, using $z = \mathrm{i}s$, over the interval $[-1, 1]$ as

$$I(n) = \frac{\left(1+e^{2\pi i n}\right)^2}{2n}\int_{-\mathrm{i}}^{\mathrm{i}}\frac{\mathrm{d}z}{2\pi \mathrm{i}}\frac{z}{\sqrt{z^2+1}}\left(\frac{z^2+1}{(z-\lambda)(z-\bar{\lambda})}\right)^n \tag{B.5}$$

$$= \frac{\left(1+e^{2\pi i n}\right)^2}{2n}\int_{-1}^{1}\frac{\mathrm{d}s}{2\pi}\frac{\mathrm{i}s}{\sqrt{1-s^2}}\left(\frac{s^2-1}{(s+\mathrm{i}\lambda)(s+\mathrm{i}\bar{\lambda})}\right)^n\,. \tag{B.6}$$

Taking derivative with respect to $\lambda$ we obtain

$$\partial_\lambda I(n) = \frac{\left(1 + e^{2\pi i n}\right)^2}{2} \int_{-1}^{1} \frac{ds}{2\pi} \frac{s}{\sqrt{1-s^2}} \left(\frac{s^2-1}{(s+i\lambda)(s+i\bar{\lambda})}\right)^n \frac{1}{s+i\lambda}, \tag{B.7}$$

and

$$\partial_\lambda \mathcal{V}_n = 1 - \frac{1}{\pi} \int_{-1}^{1} \frac{s\,ds}{\sqrt{1-s^2}} \left(\frac{s^2-1}{(s+i\lambda)(s+i\bar{\lambda})}\right)^n \frac{1}{s+i\lambda}, \tag{B.8}$$

and

$$\mathcal{V}_n = \lambda + \bar{\lambda} - \frac{i}{\pi n} \int_{-1}^{1} \frac{s\,ds}{\sqrt{1-s^2}} \left(\frac{s^2-1}{(s+i\lambda)(s+i\bar{\lambda})}\right)^n. \tag{B.9}$$

Notice, that we make sure that the phase under the power in the integrand is always zero at $z = 0$ or $s = 0$.

To obtain the representation Eq. (43) we use the identity

$$\pi = \left(\int_{-\infty}^{-1} + \int_{-1}^{1} + \int_{1}^{\infty}\right) \frac{s\,ds}{\sqrt{1-s^2}} \left(\frac{s-1}{s+i\lambda}\right)^n \left(\frac{s+1}{s+i\bar{\lambda}}\right)^n \frac{1}{s+i\lambda}, \tag{B.10}$$

which follows from the analyticity of the integrand in the half-plane to the left of the imaginary axis for $\operatorname{Re}\lambda = \operatorname{Re}\bar{\lambda} = \nu > 0$. We rewrite Eq. (36) as

$$\partial_\lambda \mathcal{V}_n = \frac{i}{\pi} \left(\int_{-\infty}^{-1} + \int_{1}^{\infty}\right) \frac{s\,ds}{\sqrt{s^2-1}} \left(\frac{s-1}{s+i\lambda}\right)^n \left(\frac{s+1}{s+i\bar{\lambda}}\right)^n \frac{1}{s+i\lambda}. \tag{B.11}$$

The change of variables $s = 1/q$ makes the integration domain compact again. We obtain

$$\partial_\lambda \mathcal{V}_n = \frac{1}{\pi i} \int_{-1}^{1} \frac{dq}{q} \frac{1}{\sqrt{1-q^2}} \left(\frac{1+q}{1-i\lambda q}\right)^n \left(\frac{1-q}{1-i\bar{\lambda}q}\right)^n \frac{1}{1-i\lambda q}, \tag{B.12}$$

which is identical to Eq. (43). This integral is formally divergent at $q = 0$, however, the divergence is removed using the principle value prescription. This also gives $\partial_\lambda \mathcal{V}_n(0,0) = 0$ as required by parity.

Fixing the contour of integration, as in the derivation of Eq. (B.5) may result in a different branch of, generally speaking, a multi-valued function of $\lambda, \bar{\lambda}$ outside the domain $\operatorname{Re}\lambda = \operatorname{Re}\bar{\lambda} = \nu > 0$. In particular, the equivalence between Eq. (B.8) and Eq. (B.12) can be violated.

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
