# Peer review of "Emptiness Instanton in Quantum Polytropic Gas"

_SciPost Physics, doi:SciPost Phys. 18, 122 (2025)_

## Round 2 · Referee Report · Anonymous (Referee 1) · 2025-2-13

Report

This paper is a follow-up to the earlier work Ref.[5]. The main result is extending the derivation of that earlier work to explicitly cover the non-integer values of the parameter n specifying the polytropic index (calculations in Ref.[5] were only done for integer n and an analytical continuation was conjectured). The extension to the case of a continuous variable n in the present paper is done by using a technique of choosing suitable integration contours in the integral representations of solutions to the equations for the relevant instantons.

While the paper may seem a simple extension of an earlier work, I believe that the topic is sufficiently important and the result is of fundamental theoretical nature, which, in my opinion, justifies its publication in SciPost Physics. It also meets all the general acceptance criteria for publication (clarity, proper citations, etc.) as listed on the journal page. I therefore recommend the paper for publication.

There are only three minor comments that the author may consider to correct (see section "requested changes" below)

Requested changes

1- the sentence above eq.(27) mentions "in the form of series", while eq.(27) itself looks to me not a series, but a finite sum. Should it be "in the form of sums" instead?

2- an apparent typo in the caption to the left panel of Fig.3. It mentions Eqs. (31) and (33), while it should be (32) and (33), to my understanding.

3- in the footnote in p.3, as well as in the caption to Fig.2, the authors mention the Chaplygin gas with n=-1 (gamma=-1). So the reader is tempted to think that the results of the paper may somehow be analytically continued to that case, too. However, both in the figure caption and after Eq.(11) the authors specify that the analysis of the paper is only valid for gamma>1 (and thus is not valid for the Chaplygin gas). It would be helpful to the reader to clarify (possibly in the last section of the paper) if there are any implications for Chaplygin gas or for any other values gamma<1 or not. Otherwise mentioning this case and an analytic continuation in Fig.2 looks confusing.

Recommendation

Publish (easily meets expectations and criteria for this Journal; among top 50%)

---

## Round 2 · Referee Report · Anonymous (Referee 2) · 2025-2-20

Strengths

  1. The results are original and to some may appeal as elegant.
  2. The paper is well written and easy to read.

Weaknesses

  1. The results represent an incremental, if aesthetically satisfying, improvement on one of the authors' previous work.

Report

The manuscript presents an extension of one of the authors' earlier result regarding the semiclassical emptiness formation probability in a one-dimensional Galilean quantum fluid. The main result of the paper is the
proof of the formula for the emptiness formation probability in the case
of a non-integer polytropic index.
The paper is well written and is mathematically sound.
The techniques used to derive the main result may be found instructive for researchers interested in instanton calculus and one-dimensional physics.
I believe that the paper easily meets the criteria of SciPost and deserves publication without change.

Recommendation

Publish (meets expectations and criteria for this Journal)

---

## Round 2 · Referee Report · Anonymous (Referee 3) · 2025-3-3

Strengths

The result are original, and solve a nontrivial problem.
The paper is well written, and very instructive.

Weaknesses

no weaknesses

Report

The paper addresses the problem of emptiness formation in a one-dimensional polytropic gas (defined by the equation of state $P\sim\rho^\gamma$), by using a large-deviation (or instanton) calculation, and solving the related hydrodynamic equations, along the line of [24]. The paper provides an exact analytic solution to the problem for generic values of $\gamma>1$. The calculations and results extends previous ones obtained by one of the authors [5] for the discrete set of values $\gamma=1/(2n+1)$, with $n$ positive integer. The physical implications of the obtained result are discussed. The technical aspects of the derivation, although far from trivial, are explained in a very clear and instructive way. The whole paper is well and clearly written.

In conclusion, the paper fulfills completely all necessary requirements of riginality, scientific rigour, relevance, clarity, and interest.

I strongly recommend publication, modulo corrections of a few typos, see below.

Requested changes

  • Page 7, Eq (28): the last factor in numerator, $(n-m+1)$, should be corrected into $(n+m-1)$;

  • page 10, sentence before Eq (47): asymptitics should be corrected into asymptotics;

"- page 16, 1st line after Eq (60): the inline equation $w_{\pm}=u\pm c$ should be corrected into $w_{\pm}=v\pm c$.

Recommendation

Publish (easily meets expectations and criteria for this Journal; among top 50%)

---

## Round 3 · Author Response

Dear Editor,

We are grateful to you and the Referees for reviewing our manuscript and recommending its publication in SciPost. We have followed the Referees' comments and suggestions to prepare the amended version, which we are resubmitting after some minor corrections. Below are our responses to the Referees' comments:

Referee 1

1- the sentence above eq.(27) mentions "in the form of series", while eq.(27) itself looks to me not a series, but a finite sum. Should it be "in the form of sums" instead?

-- We have changed the sentence referring to the Eq. 27 as "the sum".

2- an apparent typo in the caption to the left panel of Fig.3. It mentions Eqs. (31) and (33), while it should be (32) and (33), to my understanding.

-- We have corrected the equation numbers.

3- in the footnote in p.3, as well as in the caption to Fig.2, the authors mention the Chaplygin gas with n=-1 (gamma=-1). So the reader is tempted to think that the results of the paper may somehow be analytically continued to that case, too. However, both in the figure caption and after Eq.(11) the authors specify that the analysis of the paper is only valid for gamma>1 (and thus is not valid for the Chaplygin gas). It would be helpful to the reader to clarify (possibly in the last section of the paper) if there are any implications for Chaplygin gas or for any other values gamma<1 or not. Otherwise mentioning this case and an analytic continuation in Fig.2 looks confusing.

-- We have addressed the Referee's concern about the validity range of our results and have added a sentence at the end of the first paragraph of the 'Conclusions and Open Questions' section as recommended. We believe that this way presenting some of the parameters of the Emptiness Instanton solution is justified outside the validity range, and this is clearly explained.

Referee 2 has not requests

Referee 3

  • Page 7, Eq (28): the last factor in numerator, (n−m+1) , should be corrected into (n+m−1);

  • page 10, sentence before Eq (47): asymptitics should be corrected into asymptotic;

  • page 16, 1st line after Eq (60): the inline equation w±=u±c should be corrected into w±=v±c.

-- All corrected

We believe that the current version can be published.

Sincerely yours,

AG Abanov and DM Gangardt

---

## Round 3 · List of Changes

1. The sentence above Eq 27 is changed to refer to 'the sum' instead of 'the series'.

  2. Equation numbers in the caption to Fig. 3 are corrected.

  3. A sentence 'While the solution for the Emptiness Instanton.. ' is added at the end of the first paragraph of Section 6.

  4. The factor in Eq (28) is corrected.

  5. Spelling of 'asymptotics' is corrected before Eq. (47).
  6. The inline equation after Eq. (6) is corrected.

  7. We have modified the sentence after Eq. (1) which now contains integer values of n relevant to one-dimensional gases.

  8. We added Ilya Gruzberg and Paul Krapivsky in the acknowledgments.

---

## Editorial Decision

published